# Effect of Multi-Nutrient Milk Fortification on Preterm Neonate Outcomes: A Network Meta-Analysis

**DOI:** 10.3390/nu17101651

**Published:** 2025-05-12

**Authors:** Marsha Campbell-Yeo, Courtney Gullickson, Holly McCulloch, Tim Disher, Brianna Hughes

**Affiliations:** 1School of Nursing, Dalhousie University, Halifax, NS B3H 4R2, Canada; 2Mom-Linc Lab, IWK Health, Halifax, NS B3K 6R8, Canada; holly.mcculloch@iwk.nshealth.ca; 3Faculty of Medicine, Dalhousie University, Halifax, NS B3H 4R2, Canada; courtney.gullickson@dal.ca; 4Faculty of Computer Science, Dalhousie University, Halifax, NS B3H 4R2, Canada; tim.disher@dal.ca; 5School of Nursing, University of Prince Edward Island, Charlottetown, PEI C1A 4P3, Canada; brihughes@upei.ca

**Keywords:** preterm, infant, feeding, breastmilk, bovine-based fortifier, human milk-based fortifier, network meta-analysis, outcomes

## Abstract

**Background/Objectives:** Optimal feeding regimens for preterm neonates, including the role of multi-nutrient fortification, are unknown, leading to large practice variation in comparing different feeding regimens that include fortification and their impact on outcomes for preterm infants. **Methods:** Using a network meta-analyses design, two reviewers independently extracted data. A Cochrane CENTRAL, Medline, Embase, and CINAHL search was conducted for all studies published up to 27 June 2023. Randomized clinical trials of feeding regimens for preterm infants that included multi-nutrient fortification were included. Outcomes were mortality, necrotizing enterocolitis (NEC), retinopathy of prematurity (ROP), sepsis, periventricular leukomalacia (PVL), bronchopulmonary dysplasia (BPD), time to full enteral feeds, and the Bayley II MDI developmental score. **Results**: Fifty-nine studies were included. For mortality, NEC, and time to reach full enteral feeds, the top-ranked treatment class was the mother’s own milk with donor milk and a human-milk-based fortifier. For ROP and BPD, the top-ranked treatment class was mother’s own milk with a phosphorus fortifier. For sepsis, the top-ranked treatment class was mother’s own milk with formula. For PVL, the top-ranked treatment classes were mother’s own milk and mother’s own milk with donor milk and a bovine fortifier in the two disconnected networks. For the Bayley II MDI score, the top-ranked treatment class was mother’s own milk with formula and bovine fortification. **Conclusions:** Treatment rankings are consistent with the underlying hypothesis that increased mother’s own milk intake appears to be associated with better clinical outcomes. This review provides the first global view of interventions and highlights insufficient high-quality evidence to support or refute one fortification feeding regimen over another.

## 1. Introduction

The ideal source of nutrition for infants is mother’s own breastmilk [1]. If not available for preterm infants, mother’s own milk is often replaced or supplemented with donor breastmilk or preterm formula. Fortification of mother’s milk and donor milk is used to provide the increased protein and calorie demands that preterm infants need for growth and nutrition [2,3,4]. These multi-nutrient fortifiers are traditionally bovine-milk-based.

Evidence has led to the recommendation that preterm infants are fed mother’s own milk supplemented with donor milk when mother’s own milk is unavailable [5]. A Cochrane review suggested that when supplementation is needed, donor breastmilk resulted in a lower risk of necrotizing enterocolitis (NEC) compared to using preterm formula [6]. The evidence supporting exclusive human milk diets has led to the production of human-milk-derived milk fortifiers. Sullivan et al. compared bovine-milk- and human-milk-derived fortifiers through an unblinded randomized control trial and found a significant increase in the risk of NEC in the treatment group that received a bovine fortifier [7]. However, this group supplemented mother’s own milk with preterm formula versus the donor milk used in the comparison treatment arms. O’Connor et al. randomized preterm infants to either a bovine-derived or human-derived milk fortifier with a baseline diet of exclusive human milk (mother’s own milk supplemented with donor milk as needed). This study found no significant difference in their primary outcome of feeding intolerance as well as a dichotomous morbidity and mortality index with the use of the two fortifiers [8].

These studies provide contradictory conclusions regarding the role of human-derived milk fortification in the diet of preterm neonates. Although there is not a broad base of published comparative RCTs focusing on this topic, there are several trials including at least one treatment arm with milk fortifiers. As such, in the absence of head-to-head trials, network meta-analysis is the most robust method to compare therapies without access to individual participant data. We hypothesized that the choice of fortification practice would affect patient outcomes. As such, the aim of this study is to use a network meta-analysis methodology to synthesize the existing literature and examine the effects of multi-nutrient human-milk fortification on mortality and morbidity outcomes for preterm infants.

## 2. Methods

### 2.1. Study Design, Search Strategy, and Selection Criteria

This study was a systematic review with a network meta-analysis that followed a prespecified protocol (PROSPERO 2017: CRD42017069424) and was compliant with the Preferred Reporting Items for Systematic Reviews and Meta-Analyses (PRISMA) and Network Meta-Analyses. A database search was last updated for all studies published up to 27 June 2023. The electronic search strategy was developed in partnership with an information specialist from the Maritime SPOR SUPPORT Unit (MSSU) and included searches of the Cochrane Library Central Registry of Controlled Trials, Ovid Medline, Embase, Cumulative Index to Nursing and Allied Health Literature, and the Web of Science Core Collection. The full search strategy is available in the Appendix A. Ongoing trials were identified through ClinicalTrials.gov. No additional gray literature searching was conducted. The population of interest was preterm neonates defined as those delivered with less than 37 weeks of gestational age cared for in a neonatal intensive care unit (NICU). The intervention of interest was fortification of milk with the intention of increasing caloric or nutrient intake. Single amino acids, DHA, and probiotics supplementation alone did not qualify. To be eligible for inclusion, randomized clinical trials must have reported on at least one primary outcome and included at least one arm with human milk. Any neonatal diet was included as a comparator including exclusive formula diets.

### 2.2. Study Selection and Data Extraction

Title and abstract screening, full-text screening, and data extraction were conducted independently by pairs of reviewers using Covidence [9]. All conflicts were resolved through consensus and, if necessary, consultation with an alternative member of the review team. Data were extracted using standardized forms.

### 2.3. Outcome Measures

The primary outcomes were mortality, NEC, sepsis, bronchopulmonary dysplasia (BPD), retinopathy of prematurity (ROP), periventricular leukomalacia (PVL), time to full enteral feeds, and the Bayley II Mental Developmental Index (MDI) score. The most frequently used definition for full enteral feed from the included studies was 150 mL/kg/d, although a few of the studies include a range of 100–180 mL/kg/day. The Bayley III scores were converted to a Bayley II MDI scores using the following formula: MDI = 88.8 − (61.6 × (language composite score/100)^−1^) + (0.67 × cognitive composite score) [10].

### 2.4. Statistical Analysis

In the initial stage of our analysis, we conducted pairwise meta-analyses within a frequentist framework. Studies that had no events in any arm were excluded from these analyses. For binomial outcomes, analyses were conducted using the Mantel−Haenszel method [11] with a 0.5 continuity correction for zero events, while continuous pairwise meta-analyses were estimated using inverse variance on the mean difference scale. The DerSimonian and Laird method was utilized for random effects in both cases [12].

The network structure was explored using network diagrams with node sizes proportional to sample sizes, edge widths based on the number of studies in a pairwise comparison, and edge colors indicating aspects of the network connected exclusively through zero event studies (for binomial outcomes). Network meta-analyses were then conducted within a Bayesian framework using the modified NICE TSD 2 code to account for class effects [13]. A binomial likelihood with a logit link was used for binary variables, and a normal likelihood with an identity link was used for continuous outcomes. This approach allowed us to obtain estimates that would otherwise be disconnected or sparsely informed by borrowing strength from estimates within a given treatment class. The full JAGS code and priors are available in the Appendix A.

Informative priors were selected for treatment effects to provide shrinkage for both model stability and to more closely resemble prior beliefs for treatment effects. For example, a Normal (0,1) prior on log odds ratios implies a prior 95% confidence interval for treatment effects of 1 (0.17, 5.8) which reasonably captures the range of effect estimates estimated in large neonatal meta-analyses. Standard vague priors (e.g., 0, 100) would imply odds ratios greater than 300 are more likely than those between 1 and 5, which is unreasonable.

To address the underlying NMA assumption of transitivity, we performed a qualitative assessment of potential intransitivity by comparing study characteristics (PICO elements) across trials involved in direct and indirect comparisons. Inconsistency, the statistical manifestation of transitivity, was not assessed, despite the networks appearing well-connected, but in reality, were sparse in terms of data combined with reliance on class effect models for which the inconsistency models have not been previously developed. The de novo development and validation of these models were deemed not relevant as between-trial heterogeneity is affected by inconsistency and estimates were either very small or very imprecisely estimated. In connected networks, this suggests that inconsistency is limited if present, and in less well-connected networks, models would be underpowered to detect inconsistency if it were present. As a result, we placed greater emphasis on the qualitative assessment of intransitivity.

Both fixed and random effect models were fitted to the data, with model fit summarized in terms of absolute fit (residual deviance), deviance information criteria (DIC), and the magnitude of between-study heterogeneity. Due to the observed heterogeneity in patient populations, geography, and timing of the included studies, we a priori preferred random effects models.

### 2.5. Quality Assessment and Risk of Bias

Critical appraisals were conducted using the Cochrane risk of bias tool for RCTs [14]. Two reviewers assessed each study, with conflicts resolved through consensus or, if required, consultation with a third reviewer.

## 3. Results

### 3.1. Search Results and Study Characteristics

After the removal of duplicates, 4000 studies were screened at the title and abstract phase. At the full-text stage, 374 studies were assessed for eligibility with 315 excluded. Across the fifty-nine studies that were included in the analysis, 34 feeding interventions were identified (see Figure 1).

The 34 feeding interventions were combined into 10 classes of feeding regimens with the assumption that while not identical, the types of fortifiers assigned to each class are exchangeable. For example, the regimens with mother’s own milk and a bovine fortifier were combined once reviewing the details of the studies. The included papers were published between 1980 and 2023. A description of each included study and the risk of bias assessments are included in the Appendix A. Overall, the assessments of risk of bias were generally high due to missing results (arising from reporting biases) for each synthesis assessed.

### 3.2. Network Meta-Analysis

All league tables are included in the Appendix A.

### 3.3. Binomial Outcomes

The binomial pairwise comparisons were generally sparse, with few events and wide confidence intervals, even where pooling into classes was possible.

### 3.4. Mortality

There were 28 studies (4625 infants) included in the mortality outcome analysis. The mortality network was composed primarily of single study connects, with mother’s own milk with bovine colostrum fortification and mother’s own milk with individual fortification connected exclusively by studies with at least one zero cell. Similarly, studies assessing liquid fortifiers were both disconnected from the network and informed exclusively by zero cells. Thus, their effect was estimated entirely based on the estimated class effect for mother’s own milk with donor milk and a bovine fortifier. The fixed and random effect models have comparable absolute fit, with a preference towards the fixed effect model if both models are assumed equal a priori (0.69 vs. 0.31). The class standard deviation parameter was estimated with substantial uncertainty, with upper and lower bounds that only changed modestly from the prior. In the random effects model, the between-trial heterogeneity was estimated with large uncertainty, but with a small point estimate. This should be interpreted cautiously since direct information on heterogeneity was only informed by a small number of two study connections and a single three-study connection that consisted of studies with large uncertainty in their estimates. There is little reason to expect a fixed effect model to describe these data well given the heterogeneity in patient populations, geography, and timing of the included studies. With those limitations in mind, the small amount of estimated heterogeneity and the fact that only a minority of comparisons have multiple studies (and those estimates generally agree), and estimates from the two models are very similar.

The top-ranked treatment class was mother’s own milk with donor milk and human-milk-based (fixed effect odds ratio: 0.83 (0.25–3.26)) compared to mother’s own milk with donor milk and bovine fortification. The lowest-ranked treatment class was mother’s own milk with a bovine fortifier compared to the top-ranked treatment (fixed effect odds ratio: 0.41 (0.1–1.78)).

### 3.5. NEC

There were 35 studies (5195 infants) included in the NEC outcome analysis. Pairwise comparisons in the NEC were similar to those for mortality with generally 1 or 2 studies per analysis with wide confidence intervals, suggesting they would provide little information for random effects models. The NEC network was connected and included mixed evidence with single-, double-, and triple-study connections. Several treatments were connected exclusively through studies with zero cells (see Figure 2).

As in the mortality network, these treatments had their effects estimated primarily by the class average. Several comparisons consisted of 30–50% zero cells, and in those cases, comparisons were likely weighted primarily by studies with events in both arms. The fixed and random effects models had comparative performance with DIC favoring the fixed effect model. The random effect models showed small heterogeneity, but it was estimated with large uncertainty. Class effects were also estimated with large uncertainty that closely resembled the priors used. This was consistent with the underlying data showing evidence of sparse data to inform both treatment effects and the between trial standard deviation. Fixed and random effects models showed similar effect estimates for pairwise comparisons which were also consistent with the observed sparsity in terms of multiple study connections and general agreement between estimates from individual studies. Pairwise comparisons were generally wide, capturing meaningful benefits and harms for either treatment being compared. This uncertainty means that it was not possible to be confident about the direction of effect for all but a handful of comparisons. The top-ranked treatment class was mother’s own milk with donor milk and a human-milk-based fortifier (fixed effect odds ratio: 0.52 (0.16–1.7)) compared to mother’s milk and donor milk. The lowest-ranked treatment class was formula alone (fixed effect odds ratio: 0.2 (0.06–0.73)) compared to the top-ranked treatment.

### 3.6. ROP

There were 16 studies (1821 infants) included in the ROP outcome analysis. The ROP network consisted of a connected network with primarily single-study connections and one double-study connection. The network was also sparse in terms of arrangement, with several comparisons relying on long paths to create connected networks. Mother’s own milk was the only intervention in its class and was connected by a zero-cell study, offering little information regarding efficacy. Mother’s own milk alongside bovine colostrum was also connected by a zero cell and thus took the estimate from the overall class. Findings from the NMA of the ROP network are consistent with trends observed for mortality and NEC, where the network structure and underlying studies do not provide sufficient data to provide informative estimates of comparative efficacy. Credible intervals for both the class and between-study standard deviations are wide, reflecting the small amount of information provided for estimation of either. Both the fixed and random effect models provide estimates for some comparisons which lack credibility. This marked uncertainty likely also explains the unexpected finding of human-derived fortifiers being the lowest-ranked therapy and phosphorus being ranked as the best. The top-ranked treatment class was mother’s own milk with a phosphorus fortifier (fixed effect odds ratio: 0.93 (0.13–5.93)) compared to mother’s own milk and bovine fortification. The lowest-ranked treated was mother’s own milk with donor milk and human fortification (fixed effect odds ratio: 0.36 (0.08–1.74)) compared to the top-ranked treatment.

### 3.7. Sepsis

There were 36 studies (4855 infants) included in the sepsis outcome analysis. The sepsis network consisted of two distinct networks that shared common classes allowing them to be connected through a class effect assumption. The network consisted of a mix of single-, double-, and triple-study collections with mixed evidence. Only the treatment arm with mother’s own milk combined with donor milk and a liquid bovine fortifier was connected exclusively through a zero-cell study. The random effects model provided a small improvement in absolute model fit which was mostly removed by the penalization of model complexity in DIC, leading to approximately equal weights for both models. The estimation of the class standard deviation remained uncertain although the between-trial heterogeneity parameter was estimated with more precision than either NEC or ROP. The top-ranked treatment was mother’s own milk with formula (random effects odds ratio: 0.99 (0.39–2.39)) compared to mother’s own milk. The lowest-ranked was formula alone (random effects odds ratio: 0.42 (0.17–1.03)) compared to the top-ranked treatment.

### 3.8. PVL

There were 7 studies (882 infants) included in the PVL outcome analysis. Limitations of the PVL network were like those of previously highlighted binomial events. These events were generally rare, and effect estimates captured appreciable benefits and harms in most cases. The PVL network consisted of two distinct networks without any shared classes. These networks were analyzed separately. The network consisted of single- and double-study connections. As with earlier networks, a fixed effect model was preferred, but the class and between-trial SDs were imprecisely estimated in both sub networks. This led to large uncertainty in the estimated effects. The top-ranked treatment in the first network was mother’s own milk (fixed effect odds ratio: 0.91 (0.14–6.84)) compared to formula alone. The lowest-ranked was mother’s own milk with formula (fixed effect odds ratio: 0.61 (0.21–2.18)) compared to the top-ranked treatment. In the second network, the top-ranked treatment was mother’s own milk with donor milk and a bovine fortifier compared to mother’s own milk with donor milk and a human fortifier (fixed effect odds ratio: 0.94 (0.3–3.18)).

### 3.9. BPD

There were 20 studies (2405 infants) included in the BPD outcome analysis. As with other outcomes, connections were primarily single-study with mother’s own milk and a bovine fortifier and colostrum connected via an arm with zero-events, leaving its estimates to be derived from other treatments in the class. Powdered fortifiers and individual fortification cannot be connected to the network without additional assumptions, and liquid fortifiers relied on class effects to connect. The fixed effect model was preferred, but the estimated heterogeneity in the random effects model was small to moderate. Parameters for between-trial heterogeneity and within-class standard deviation were generally well estimated. Interpretation of pairwise comparisons was the same between models, with the sole difference being slightly wider credible intervals. The top-ranked was mother’s own milk with a phosphorus fortification (fixed effect odds ratio: 0.78 (0.24–2.24)) compared to the second-ranked position of mother’s own milk with donor milk and a human fortifier. The lowest-ranked class compared to phosphorus was mother’s own milk with donor milk (fixed effect odds ratio: 0.38 (0.11–1.25)).

### 3.10. Continuous Outcomes

Pairwise comparisons of continuous outcomes suffer from many of the same limitations of those from the binomial network. The networks were sparse and informed by comparisons with large uncertainty compared to the treatment effects and therefore offer limited information on between-study heterogeneity, suggesting fixed effect models may be more appropriate. The lack of evidence of heterogeneity should not be taken as evidence of no heterogeneity, and fixed effect models should be interpreted knowing that their estimates may be unstable as evidence in this area continuous to accrue. These analyses relied on a comparison of studies conducted more than twenty years apart in different geographical locations and thus compare babies with very different background care and post-discharge developmental support. Lastly, the sparseness of the network combined with the large uncertainty of estimated effects was likely to lead to treatment rankings and comparative estimates that may not be congruent with biological expectations.

#### 3.10.1. Time to Full Enteral Feeds

There were 16 studies (2158 infants) included in the time to full enteral feeds analysis. Results from pairwise meta-analysis of time to full enteral feed present similar challenges to those from binomial outcomes. Only two comparisons have more than a single study. While there was some heterogeneity in estimates for mothers’ own milk with donor milk and a powdered bovine fortifier, the small number of studies combined with the relative imprecision of the effect estimates means that heterogeneity was estimated with substantial uncertainty. The highest-ranked class was mother’s own milk with donor milk and a human fortifier compared to mother’s own milk with formula (fixed effect mean difference: −0.55 (−5.73 to 4.49)). The lowest rank compared to mother’s own milk with donor milk and a human fortifier was formula (fixed effect mean difference: −4.98 (−9.75 to −0.18)).

#### 3.10.2. Bayley II MDI

There were 6 studies (1233 infants) included in the time to Bayley II MDI analysis. Comparisons were all based on single-study connections of studies with large uncertainty relative to the estimated effect. Some point estimates were inconsistent with the prior expectation based on biological rational. For example, mothers’ own milk with donor milk and a bovine fortifier was worse compared to mothers’ own milk with preterm formula and a bovine fortifier. However, this may be an artifact of the uncertainty of the estimates. The highest-ranked class was mother’s own milk with formula and a bovine fortification compared to the second-ranked class of formula (fixed effect mean difference: 0.81 (−7.12 to 8.88)). The lowest-ranked class was mother’s own milk with donor milk and human fortification compared to mother’s own milk with formula and bovine fortification (fixed effect mean difference: 3.45 (−5.18 to 12.11)).

#### 3.10.3. Exploration for Inconsistency

As previously included in the methods, statistical inconsistency was not assessed at this time since data were so sparse and reliant on class effect assumptions for which were not developed and tested.

## 4. Discussion

We aimed to synthesize and compare different feeding regimens that include fortification on mortality and morbidity outcomes for preterm infants. For most outcomes, treatment rankings are consistent with the underlying hypothesis that increased human milk intake is associated with better clinical outcomes.

Findings from the network analyses demonstrated that there is limited empirical evidence to prefer a specific fortification strategy over any of the alternatives. Pairwise comparisons of the existing evidence are wide, which creates uncertainty about even the direction of effect for some comparisons. Across comparisons, networks were sparse in terms of both events for binomial outcomes and the number of studies contributing to any direct comparison. The studies included in this review and network do not provide sufficient data to create informative estimates of the preferred fortification strategy for preterm infants.

The network quality is impacted by many single-study connections. The variance of indirect comparisons increases as the chain between target comparators grows and ranks themselves are already estimated with large uncertainty. Further, there are theoretical reasons to believe that sparsely connected networks are less reliable and more likely to result in inconsistent or highly variable findings. Our findings are consistent with a recent review examining the quality of 27 national or international nutrition guidelines reporting an insufficient body of quality evidence to recommend optimal fortification practices for preterm infants [15].

A relevant clinical outcome that is not included in this review is feed tolerance. Findings related to this outcome appear mixed. Kotha et al. compared human- and bovine-based fortifiers in a small RCT with 25 very-low-birthweight infants and suggested the human milk group had a lower risk of feed intolerance [16]. In an RCT including 112 very-low-birthweight neonates comparing expressed breast milk (EBM) alone and EBM combined with a human-milk-based fortifier (HMF), infants receiving an HMF showed a significantly better weight gain and a head circumference at 6 weeks and at discharge. However, they had a higher reported feeding intolerance (25.7% compared to 10.5% in the EBM alone) [17].

The clinical significance of feeding tolerance on reducing NEC and its effect on growth have not been clearly demonstrated. Given the higher cost of human-based fortifiers, a clear reduction in NEC would be needed to show an economical advantage. Like other reviews in this area, we found a lack of high-quality RCTs published [18]. The network did support potential benefits of the reduction in NEC with the human-based products, but the same issues of data quality are present for this outcome.

Uthaya et al. published an open-label RCT comparing an exclusive human milk diet versus one that contained bovine milk products with a focus on growth outcomes [19]. Due to the reported outcomes, this study was not eligible for this review. However, they found no differences in feed tolerance, days of parenteral nutrition, exclusive breastfeeding at discharge, length of stay, or body composition. Gates et al. recently looked at a new sterile, human-based fortifier [20]. Due to the reported outcomes, it was also not eligible for this review. Like Uthaya et al. [19], although well tolerated, there was no significant difference in growth outcomes noted.

To our knowledge, no clinical trials focused specifically on preterm infant fortification practices have been reported since we completed our search. In one clinical trial comparing unfortified donor milk versus infant formula on the outcomes of extremely-low-birth-weight infants, which was conducted following the search dates for this paper, their results were similar to our reported findings [21]. In a study in the US, 483 infants were randomized, with 239 receiving donor milk and 244 receiving preterm formula. The median gestational age was 26 weeks (IQR: 25–27 weeks), and the median birth weight was 840 g (IQR: 676–986 g). There was no significant difference in raw or adjusted neurodevelopmental (BSID scores) and motor outcomes measured between 22 and 26 months. There was a lower incidence of NEC (4.2% vs. 9.0%) in the human milk group compared to in the formula group [21].

The main strength of the network meta-analysis was that a broad range of studies could be included in the analysis. In comparison to other published systematic reviews on this topic, including the Cochrane reviews [22,23], we were able to include many more RCTs. However, this broad inclusion introduces variability in terms of the study protocols.

The finding of small amounts of heterogeneity (and by extension minimal evidence of inconsistency) is surprising given the differences in gestational age, geographic region, and time. Each of these would be expected to potentially be an effect modifier. Infants of lower gestational age are at higher risk for increased morbidity and mortality, and therapy could be reasonably expected to have a larger effect on subsequent development. It would be reasonable to expect that therapies that are more aggravating to neonatal digestion may result in feeding intolerance in older infants but potentially more severe consequences in those of younger gestational age. The issue of baseline risk is also potentially problematic since trials conducted in lower-gestational-age infants are expected to have higher rates of any events. This can be partially addressed through an assumption of odds ratios being invariant to baseline risk but can still raise problems regarding bias related to rare events. The geographic region can be a potential effect modifier since there is well-established heterogeneity in neonatal care within and between countries as well as subsequent developmental outcomes. Again, it may be possible that therapies can appear better or worse as a result of the underlying practices related directly either to feeding (e.g., use of TPN and speed to full feeds) or to other aspects of care. If the risk under a comparator therapy is already low, it may obscure additional benefits of a more supportive therapy. Practices are also closely related to time, which introduces an additional layer of uncertainty regarding variation within and between countries over the span of a 20–30-year period.

Overall, we would expect some degree of intransitivity. While lack of heterogeneity makes us feel a little more confident with our findings, we acknowledge that it does not guarantee a lack of inconsistency.

The requirement to use strong priors on treatment effects can be seen as a limitation since typically priors are set to be none or minimally informative. The type of priors used in this analysis was considered to provide appropriate support for a range of treatment effects that would be comparable with some of the most effective therapies identified in neonatology. Setting priors in this way is a recommended best practice and ensures that analyses are not inappropriately driven by extreme results from small studies or those with rare events. These priors also provide a better estimate of true prior belief in comparisons and therefore allows the interpretation of results to benefit from the use of a full Bayesian framework, which is valuable when many comparisons are made since credible intervals are calibrated under the priors used, thus reducing issues of multiplicity.

Additional limitations were that we did not differentiate between targeted, adjustable, and standard fortification methods. In addition, the timing of fortification emerged as a variable of interest in terms of clinical outcomes, and it was highly variable between the included studies. Another variable that has been studied is the potential difference between donor milk and mother’s own milk on developmental outcomes, for example in [24]. These elements were not considered individually in this review. The timeline of the studies from 1980 to 2023 included many other changes in care protocols other than feeding that were likely to also influence these clinical outcomes. Lastly, limitations related to the networks were as follows: we relied on class assumptions for connectivity; all interventions were not able to be reported across all outcomes based on the current literature; there were high levels of zero cells in some outcomes (e.g., NEC); and the incidence of mostly single-study connections prevented estimation of heterogeneity/exploration of inconsistency.

Nevertheless, this systematic review and network meta-analysis are the broadest analysis of the evidence for the preferred fortification method for preterm infants. Treatment rankings are consistent with the underlying hypothesis that higher intake of human milk, most notably mother’s own milk, appears to be associated with better clinical outcomes. However, there remains insufficient high-quality evidence to support or refute one fortification feeding regimen over another. Future studies could include collaboration across multiple research teams to conduct an individual patient data meta-analysis to allow for finer adjustment for possible effect-modifying variables that cannot be controlled for at the population level. Comparative studies using existing national databases linked with neonatal follow-up programs may provide insights into neurodevelopmental outcomes as this approach may allow for larger numbers of included infants.

## Figures and Tables

**Figure 1 nutrients-17-01651-f001:**
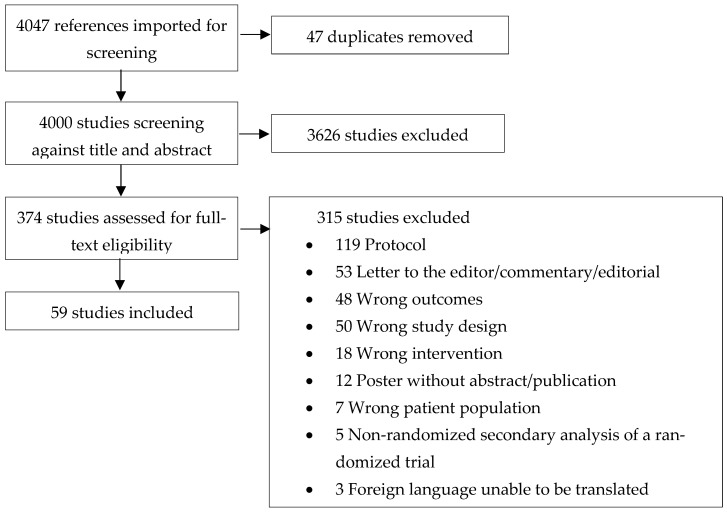
PRISMA flow diagram.

**Figure 2 nutrients-17-01651-f002:**
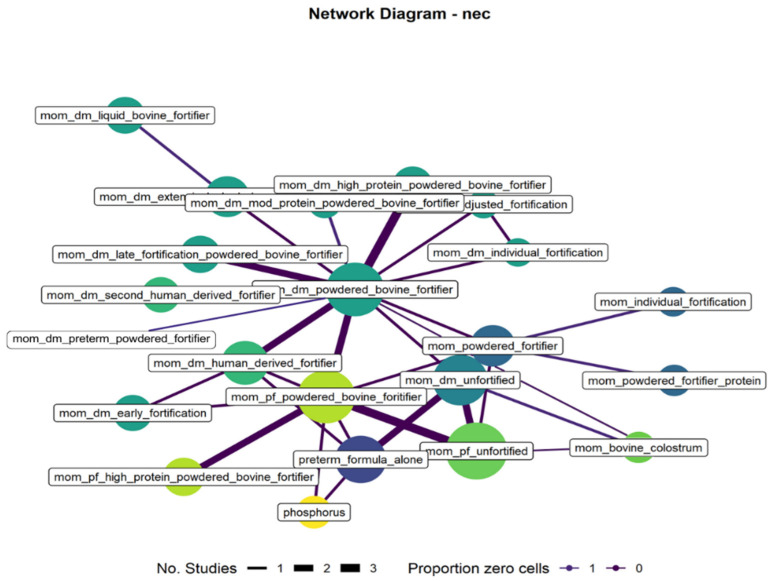
NEC network plot.

## Data Availability

The original contributions presented in this study are included in the article or Appendix A. Further inquiries can be directed to the corresponding author.

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
