# Peer review of "Effect of Multi-Nutrient Milk Fortification on Preterm Neonate Outcomes: A Network Meta-Analysis"

_nutrients, 2025, doi:10.3390/nu17101651_

Round 1
Reviewer 1 Report
Comments and Suggestions for Authors
Dear Editors,
I'm greatful for opportunity to review astricle entitled: Effect of multi-nutrient milk fortification on preterm neonate outcomes: a network meta-analysis
Study presents a valuable contribution to the understanding of optimal feeding regimens in preterm neonates, particularly highlighting the importance of multi-nutrient fortification. The structured approach using a network meta-analysis is commendable, and the comprehensive search strategy for randomized clinical trials strengthens the validity of findings.
The results present clear rankings of various treatment classes based on significant outcomes, which is particularly useful for clinicians making feeding decisions in neonatal care. The emphasis on mother’s own milk, which is consistently associated with better clinical outcomes, reflects a growing consensus in pediatric medicine and addresses a crucial aspect of neonatal nutrition.
However, there are a few areas for potential improvement:
While the methodology of the network meta-analysis is mentioned, further details on inclusion criteria, how data was extracted, and the which tools were used for assessment would enhance transparency.
Enlarge section that discusses limitations in the studies reviewed or in methodology would provide a more balanced perspective.
It would be beneficial to suggest specific areas for future research, such as gaps in data or potential longitudinal studies to follow up on the developmental outcomes assessed by the Bayley II MDI score.
Including demographic data about the populations in studies (e.g., birth weights, gestational ages) could provide insight into the generalizability of results.
Explore how feeding regimens impact other factors like parental involvement, cost-effectiveness, and length of NICU stay, as these are also critical components in neonatal care.
Author Response
Dear Editors,
I'm grateful for opportunity to review article entitled: Effect of multi-nutrient milk fortification on preterm neonate outcomes: a network meta-analysis
Study presents a valuable contribution to the understanding of optimal feeding regimens in preterm neonates, particularly highlighting the importance of multi-nutrient fortification. The structured approach using a network meta-analysis is commendable, and the comprehensive search strategy for randomized clinical trials strengthens the validity of findings.
The results present clear rankings of various treatment classes based on significant outcomes, which is particularly useful for clinicians making feeding decisions in neonatal care. The emphasis on mother’s own milk, which is consistently associated with better clinical outcomes, reflects a growing consensus in pediatric medicine and addresses a crucial aspect of neonatal nutrition.
However, there are a few areas for potential improvement:
While the methodology of the network meta-analysis is mentioned, further details on inclusion criteria, how data was extracted, and the which tools were used for assessment would enhance transparency.
- Thank you, the following information that addresses this comment can be found on page 4 and 5 and 7”.
“The population of interest was preterm neonates defined as those delivered less than 37 weeks gestational age cared for in a neonatal intensive care unit (NICU). The intervention of interest was fortification of milk with the intention of increasing caloric or nutrient intake. Single amino acids, DHA, and probiotics supplementation alone did not qualify. To be eligible for inclusion, trials must have reported on at least one primary outcome and included at least one arm with human milk. Any neonatal diet was included as a comparator including exclusive formula diets.
2.2 Study selection and data extraction
Title and abstract screening, full-text screening, and data extraction were conducted independently by pairs of reviewers using Covidence.[9] All conflicts were resolved through consensus and, if necessary, consultation with an alternative member of the review team. Data were extracted using standardized forms.
2.3 Outcome measures
The primary outcomes were mortality, NEC, sepsis, bronchopulmonary dysplasia (BPD), retinopathy of prematurity (ROP), periventricular leukomalacia (PVL), time to full enteral feeds, and the Bayley II Mental Developmental Index (MDI) score. Each of the studies defined full enteral feed differently, with a range from 100-180 mL/kg/d. The most frequently used definition for full enteral feed from the included studies was 150 mL/kg/d. The Bayley III scores were converted to a Bayley II MDI scores using the following formula: MDI = 88.8 - (61.6 × (language composite score/100)-1) + (0.67 × cognitive composite score).
…2.5 Quality assessment and risk of bias
Critical appraisals were conducted using the Cochrane risk of bias tool for RCTs. [14] Two reviewers assessed each study, with conflicts resolved through consensus or, if required, consultation with a third reviewer.”
Enlarge section that discusses limitations in the studies reviewed or in methodology would provide a more balanced perspective.
- Thank you, we have revised the discussion to distinguish more clearly and expand on the study limitations.
It would be beneficial to suggest specific areas for future research, such as gaps in data or potential longitudinal studies to follow up on the developmental outcomes assessed by the Bayley II MDI score.
- Thank you, we have included this information in the conclusion.
Including demographic data about the populations in studies (e.g., birth weights, gestational ages) could provide insight into the generalizability of results.
- We have included detailed information regarding study characteristics in supplemental Table S2.2. We would be happy to move this information to the main manuscript if the editors prefer.
Explore how feeding regimens impact other factors like parental involvement, cost-effectiveness, and length of NICU stay, as these are also critical components in neonatal care.
- Given the degree of additional information added to the manuscript to address other reviewer comments that were directly aligned with the objective of the paper, we elected not to add further information regarding this point that while we absolutely agree is important, we did not believe we could robustly address within the word limit constraints at this time. We are certainly willing to do so at the editors discretion regarding manuscript length.
Reviewer 2 Report
Comments and Suggestions for Authors
The review paper titled “nutrients-3552039_Effect of multi-nutrient milk fortification on preterm neonate outcomes: a network meta-analysis” is submitted to the “Pediatric Nutrition” section of this journal. It is presented as a systematic review aiming to utilise network meta-analysis methodology to synthesise existing literature and assess the impact of multinutrient human milk fortification on mortality and morbidity outcomes in preterm infants. However, as indicated in the manuscript, it should be explicitly classified as a meta-analysis review.
Comments:
1.-Clarification of Study Design
The title suggests that the study is a network meta-analysis (NMA). However, in both the introduction and methodology sections, the manuscript refers to it as a meta-analysis. This inconsistency should be addressed. A network meta-analysis (NMA) is a statistical technique used to compare multiple treatments simultaneously, even in the absence of direct head-to-head comparisons. It extends traditional meta-analysis by incorporating both direct and indirect evidence. This approach is not commonly employed in nutritional epidemiology, and its relevance to the study should be clarified.
2-Abstract
The time period covered in the literature review is only indicated as “conducted to June 2023.” The exact timeframe for the literature search should be explicitly stated, as this is crucial for comparison with other reviews.
The number of studies included in the review should be specified.
The methodology employed should be clearly outlined.
The results should be supported by statistical data.
The conclusion should directly address the study’s primary objective.
Line 146 states that the study covers 1980 to 2023, yet we are currently in 2025. This needs verification.
3.-Introduction
The introduction discusses the importance of human milk fortification, but the underlying hypothesis should be more explicitly stated, particularly given the extensive literature on this topic.
4.-Methodology
The study is described as a systematic review with a network meta-analysis. However, this classification should be consistently used throughout the manuscript, including the title.
The review period is not clearly specified, with only “updated in June 2023” mentioned. Given that the study is being submitted in 2025, this should be updated.
Line 71 states that ongoing trials were identified via ClinicalTrials.gov, but Line 56 asserts that there is significant feeding literature with treatment arms using milk fortifiers. This discrepancy should be clarified.
Section 2.2 (Selection and Data Extraction) states that Covidence was used, but the inclusion and exclusion criteria applied through this tool are not specified.
A PRISMA flow diagram illustrating article selection should be included in the methodology section, rather than in the results section.
Lines 118–126 should be further elaborated on in the discussion, particularly regarding the stability of the reported results.
Results
The PRISMA diagram states that 59 studies were included, yet Line 142 describes 34 feeding interventions.
Line 182 mentions 35 studies included in the NEC outcome analysis, while Line 209 refers to 16 studies in the ROP outcome analysis, totalling 63 studies. The discrepancy in study numbers must be clarified.
Section 3.2 (Network Meta-Analysis) should include:
Indirect Comparisons between studies.
Network of Treatments to demonstrate how all interventions were connected.
Ranking of Treatments based on efficacy.
Bayesian or Frequentist Approach used for statistical modelling.
5.-Results should be supported by figures and tables for improved reader comprehension.
Each subsection should indicate the number of studies analysed and reference a table summarising key findings.
Discussion
The methodological quality and risk of bias of the included studies should be assessed, as is standard for systematic reviews.
If a meta-analysis was conducted, results should be presented more consistently, with supporting figures and tables.
6.-The discussion frequently addresses the presence or absence of clinical trials, but this is not clearly structured within the manuscript.
7.-Conclusion
The conclusion should clearly state the study’s contribution based on the results obtained.
Overall Evaluation
This is an interesting and relevant study; however, several key methodological and conceptual aspects require clarification. The manuscript would benefit from greater consistency in terminology, clearer methodological explanations, and additional figures and tables appropriate for both meta-analysis and network meta-analysis studies.
Author Response
Dear Editors,
The review paper titled “nutrients-3552039_Effect of multi-nutrient milk fortification on preterm neonate outcomes: a network meta-analysis” is submitted to the “Pediatric Nutrition” section of this journal. It is presented as a systematic review aiming to utilise network meta-analysis methodology to synthesise existing literature and assess the impact of multinutrient human milk fortification on mortality and morbidity outcomes in preterm infants. However, as indicated in the manuscript, it should be explicitly classified as a meta-analysis review.
Comments:
1.-Clarification of Study Design
The title suggests that the study is a network meta-analysis (NMA). However, in both the introduction and methodology sections, the manuscript refers to it as a meta-analysis. This inconsistency should be addressed. A network meta-analysis (NMA) is a statistical technique used to compare multiple treatments simultaneously, even in the absence of direct head-to-head comparisons. It extends traditional meta-analysis by incorporating both direct and indirect evidence. This approach is not commonly employed in nutritional epidemiology, and its relevance to the study should be clarified.
- Thank you for your comment. We have reviewed the paper and do not see the use of the term meta-analysis in isolation of the word network with respect to the design description. As such, are unable to make any corrections in response to this comment.
- We do correctly use the term “pair-wise meta -analysis” in the initial stages of analysis (in keeping with usual analysis processes) to understand the underlying data for the analysis, and explore heterogeneity and strength of direct evidence. We believe that our statistical analysis section on page 5-7, clearly outlines the steps and processes of a network meta-analysis.
2-Abstract
The time period covered in the literature review is only indicated as “conducted to June 2023.” The exact timeframe for the literature search should be explicitly stated, as this is crucial for comparison with other reviews.
Thank you, we have added this information in both the abstract and methods sections.
The number of studies included in the review should be specified.
This has been added to the abstract.
The methodology employed should be clearly outlined.
This information is included in the abstract.
The results should be supported by statistical data.
We are happy to provide statistical data; however, give the number of outcomes, this would significantly increase the word count of the abstract. As such we included information based on the network ranking of the outcomes. We will defer to the editors if they wish us to expand the abstract word count to add statistical data for all the reported outcomes.
The conclusion should directly address the study’s primary objective.
Our objective for the study was to compare different feeding regimens that include fortification on outcomes for preterm infants using a network meta-analysis design.
Our current conclusion aligns with this objective.
Line 146 states that the study covers 1980 to 2023, yet we are currently in 2025. This needs verification.
Thank you. Exact dates have been provided.
3.-Introduction.
The introduction discusses the importance of human milk fortification, but the underlying hypothesis should be more explicitly stated, particularly given the extensive literature on this topic.
- The objective of a network meta-analysis is to compare and contrast existing evidence through both direct and indirect data sources to provide a broader overview of the evidence than a single meta-analysis. It is not typical procedure to provide a hypothesis for this type of methodologic design.
4.-Methodology
The study is described as a systematic review with a network meta-analysis. However, this classification should be consistently used throughout the manuscript, including the title.
- We are uncertain what changes the reviewer asks us to make regarding the title which is currently:
“Effect of multi-nutrient milk fortification on preterm neonate outcomes: a network meta-analysis”
The review period is not clearly specified, with only “updated in June 2023” mentioned. Given that the study is being submitted in 2025, this should be updated.
We have added the exact dates of the review. We have also included information in the discussion regarding lack of trials reported following our search.
Line 71 states that ongoing trials were identified via ClinicalTrials.gov, but Line 56 asserts that there is significant feeding literature with treatment arms using milk fortifiers. This discrepancy should be clarified.
We apologize that this statement was not clear. We have revised to improve clarity.
Section 2.2 (Selection and Data Extraction) states that Covidence was used, but the inclusion and exclusion criteria applied through this tool are not specified.
A PRISMA flow diagram illustrating article selection should be included in the methodology section, rather than in the results section.
For the purpose of a systematic search conducted as part of a network meta-analysis, the completed PRISM demonstrating the included studies is typically part of the results section. https://static1.squarespace.com/static/65b880e13b6ca75573dfe217/t/65b9e1688e834d1c620292b5/1706680681041/PRISMA+NMA+checklist.pdf
Lines 118–126 should be further elaborated on in the discussion, particularly regarding the stability of the reported results.
We have added additional information as per a prior reviewer’s comment regarding study limitations.
Results
The PRISMA diagram states that 59 studies were included, yet Line 142 describes 34 feeding interventions.
- Thank you, we have clarified that across the 59 included studies, that 34 different feeding interventions were identified.
Line 182 mentions 35 studies included in the NEC outcome analysis, while Line 209 refers to 16 studies in the ROP outcome analysis, totalling 63 studies. The discrepancy in study numbers must be clarified.
- Thank you for your comment, however, we are uncertain where you see 63 studies, as this is not mentioned in the paper? To clarify, from our 59 included studies, we report on how many report on the specific outcomes of interest.
Section 3.2 (Network Meta-Analysis) should include:
Indirect Comparisons between studies.
Network of Treatments to demonstrate how all interventions were connected.
Ranking of Treatments based on efficacy.
Bayesian or Frequentist Approach used for statistical modelling.
- We have included all of the above information in the supplementary materials. We are happy to discuss with the editors if they prefer some or all to be moved to the main manuscript based on word count limitations.
5.Results should be supported by figures and tables for improved reader comprehension.
Each subsection should indicate the number of studies analysed and reference a table summarising key findings.
- We have included number of included studies under each outcome in the main paper and have included tables in the supplementary materials. We are happy to discuss with the editors if they prefer some or all of the tables to be moved to the main manuscript.
Discussion
The methodological quality and risk of bias of the included studies should be assessed, as is standard for systematic reviews.
- We have included details for the assessment of risk of bias in the supplementary materials. We are happy to discuss with the editors if they prefer some or all to be moved to the main manuscript.
If a meta-analysis was conducted, results should be presented more consistently, with supporting figures and tables.
- We have included this information in the supplementary materials. We are happy to discuss with the editors if they prefer some or all to be moved to the main manuscript.
6.-The discussion frequently addresses the presence or absence of clinical trials, but this is not clearly structured within the manuscript.
- We have clarified this in the discussion.
7.-Conclusion
The conclusion should clearly state the study’s contribution based on the results obtained.
-Our study is the first to provide a global overview of feeding interventions comparing both direct and indirect data. Our findings highlight that lack of strong evidence to guide clinical decision-making concerning optimal feeding practices in preterm infants.
Overall Evaluation
This is an interesting and relevant study; however, several key methodological and conceptual aspects require clarification. The manuscript would benefit from greater consistency in terminology, clearer methodological explanations, and additional figures and tables appropriate for both meta-analysis and network meta-analysis studies.
The manuscript addresses an important and clinically relevant topic regarding the impact of multi-nutrient milk fortification on preterm neonate outcomes. The authors employ a systematic review and network meta-analysis approach to synthesize existing literature on the subject. The study is well-structured and follows appropriate methodological guidelines, including compliance with PRISMA and the use of PROSPERO registration.
However, despite the well-executed methodology, the findings are limited by the sparse data network, wide confidence intervals in pairwise comparisons, and reliance on many single-study connections. This results in considerable uncertainty regarding the preferred fortification strategy. While the study does indicate potential benefits of increased human milk intake, it does not provide conclusive evidence to support one specific fortification approach over another.
Reviewer 3 Report
Comments and Suggestions for Authors
The manuscript addresses an important and clinically relevant topic regarding the impact of multi-nutrient milk fortification on preterm neonate outcomes. The authors employ a systematic review and network meta-analysis approach to synthesize existing literature on the subject. The study is well-structured and follows appropriate methodological guidelines, including compliance with PRISMA and the use of PROSPERO registration.
However, despite the well-executed methodology, the findings are limited by the sparse data network, wide confidence intervals in pairwise comparisons, and reliance on many single-study connections. This results in considerable uncertainty regarding the preferred fortification strategy. While the study does indicate potential benefits of increased human milk intake, it does not provide conclusive evidence to support one specific fortification approach over another.
- Justification for Network Meta-Analysis: The rationale for conducting a network meta-analysis instead of a traditional pairwise meta-analysis is briefly mentioned but not fully justified. While the authors state that there are limited direct randomized controlled trials (RCTs) comparing human-derived and bovine-derived fortifiers, more explanation on why a network meta-analysis is the best approach for synthesizing the data is needed.
- Heterogeneity and Inconsistency Assessment: The manuscript mentions that inconsistency was not assessed due to data sparsity and reliance on class-effect models. However, inconsistency is a key assumption in network meta-analysis, and its absence can affect the validity of conclusions. A sensitivity analysis or at least a qualitative discussion of potential inconsistency sources should be included.
- Selection of Informative Priors in Bayesian Analysis: The authors justify their choice of informative priors in the Bayesian framework. However, further details on prior sensitivity analysis would strengthen the credibility of the results. How would different prior distributions impact the estimates?
- Definition of Full Enteral Feeding: The definition of full enteral feeding varies across studies (100-180 mL/kg/day). While the most common cutoff was 150 mL/kg/day, how was this variability accounted for in the meta-analysis? Was a subgroup analysis conducted based on different definitions?
- Impact of Study Design Differences on Results: Some included studies had different baseline diets (e.g., use of donor milk vs. preterm formula as supplementation). This could introduce significant heterogeneity. Was any stratified analysis conducted to address this potential confounder?
Author Response
Dear Editors,
- Justification for Network Meta-Analysis: The rationale for conducting a network meta-analysis instead of a traditional pairwise meta-analysis is briefly mentioned but not fully justified. While the authors state that there are limited direct randomized controlled trials (RCTs) comparing human-derived and bovine-derived fortifiers, more explanation on why a network meta-analysis is the best approach for synthesizing the data is needed.
- Thank you, we have provided additional justification in the introduction.
Heterogeneity and Inconsistency Assessment: The manuscript mentions that inconsistency was not assessed due to data sparsity and reliance on class-effect models. However, inconsistency is a key assumption in network meta-analysis, and its absence can affect the validity of conclusions. A sensitivity analysis or at least a qualitative discussion of potential inconsistency sources should be included.
We thank the reviewer for raising this important point regarding the assessment of inconsistency. We acknowledge that inconsistency is a critical consideration in network meta-analysis.
We wish to clarify the distinction between the underlying NMA assumption of transitivity and its statistical manifestation, inconsistency. As stated in our submission, we performed a qualitative assessment of potential intransitivity by comparing study characteristics (PICO elements) across trials involved in direct and indirect comparisons. This qualitative assessment addresses the fundamental assumption directly.
Regarding the statistical assessment of inconsistency, our manuscript noted this was not performed due to data sparsity and the use of class-effect models, for which validated inconsistency assessment methods are not readily available. Furthermore, we believe our assessment of between-trial heterogeneity provides an indirect evaluation relevant to this issue. As the reviewer knows, inconsistency and heterogeneity are closely related. Our findings showed that heterogeneity was either minimal (suggesting limited inconsistency, if any) or very imprecisely estimated (indicating that any statistical test for inconsistency would be severely underpowered). Given these findings,
To improve clarity for the reader, we will revise the relevant section of the manuscript to explicitly state the distinction between transitivity (assessed qualitatively) and inconsistency (statistical manifestation), and to better explain how our interpretation of heterogeneity provides an indirect check, acknowledging the limitations.
The finding of small amounts of heterogeneity (and by extension minimal evidence of inconsistency) is surprising given the differences in gestational age, geographic region, and time. Each of these would be expected to potentially be an effect modifier. Infants of lower gestational age are at higher risk for increased morbidity and mortality and therapy could be reasonably expected to have a larger effect on subsequent development. It would be reasonable to expect that therapies that are more aggravating to neonatal digestion may result in feeding intolerance in older infants but potentially more severe consequences in those of younger gestational age. The issue of baseline risk is also potentially problematic since trials conducted in lower gestational age infants are expected to have higher rates of any events. This can be partially addressed through an assumption of odds ratios being invariant to baseline risk, but can still raise problems regarding bias related to rare events. Geographic region can be potentially effect modifying since there is well established heterogeneity in neonatal care within and between countries as well as subsequent developmental outcomes. Again, it may be possible that therapies can appear better or worse as a result of the underlying practices either related directly to feeding (eg use of TPN, speed to full feeds) or to other aspects of care. If the risk under a comparator therapy is already low, it may obscure additional benefit of a more supportive therapy. Practices are also closely related to time, which introduces an additional layer of uncertainty regarding variation within and between countries over the span of a 20-30 year period.
Overall, we would expect some degree of intransitivity, and while lack of heterogeneity makes us feel a little more confident with our findings, we acknowledge that it doesn't guarantee a lack of inconsistency.
- Selection of Informative Priors in Bayesian Analysis: The authors justify their choice of informative priors in the Bayesian framework. However, further details on prior sensitivity analysis would strengthen the credibility of the results. How would different prior distributions impact the estimates?
Thank you, we have recognized the importance of these priors in the analysis but have not conducted sensitivity analyses given the alternative typically vague priors would create issues for convergence and magnify issues related to multiple testing and sparse networks. Given the number of treatments and lack of power in comparisons use of vague priors would have a high probability of generating implausibly large treatment effects with spuriously high confidence. For that reason we agree with Efthimiou and Barrientos that more informative priors are more credible in these situations. For those reasons we don’t believe it to be a reasonable sensitivity analysis, since in the event of any incoherence we would always find the current results to be more credible._Sensitivity analysis critieria:_https://bmcmedresmethodol.biomedcentral.com/articles/10.1186/1471-2288-14-11 __Multiple testing in NMA:_https://onlinelibrary.wiley.com/doi/full/10.1002/jrsm.1377 _https://pubmed.ncbi.nlm.nih.gov/39552751/ _
We have added additional information in the manuscript to further elucidate this point.
“The requirement to use strong priors on treatment effects can be seen as a limitation since typically priors are set to be none or minimally informative. The type of priors used in this analysis were considered to provide appropriate support for a range of treatment effects that would be comparable with some of the most effective therapies ever identified in neonatology. Setting priors in this way is a recommended best practice and ensures that analyses are not inappropriately driven by extreme results from small studies or those with rare events. These priors also provide a better estimate of true prior belief in comparisons and therefore allows the interpretation of results to benefit from being fully Bayesian which is valuable when many comparisons are being made since credible intervals are calibrated under the priors used and thus reduce issues of multiplicity.”
- Definition of Full Enteral Feeding: The definition of full enteral feeding varies across studies (100-180 mL/kg/day). While the most common cutoff was 150 mL/kg/day, how was this variability accounted for in the meta-analysis? Was a subgroup analysis conducted based on different definitions?
Subgroup analyses are not feasible given the network structure. As such we relied on the relative effects being constant over the range. We have captured this variation in our qualitative analysis of heterogeneity/intransitivity and have included as a limitation.
The majority of studies had a cutoff of 150ml.kg.d. While the final interpretation regarding implications of including a range is a clinically based one – specific considerations underpinning the interpretation would be: 1. Is the range of 100-180 concerning in terms of relative risk of, e.g. NEC remaining constant? It’s fine for the baseline risk of the event to change e.g. more aggressive feeding might be more risky (absolute values change) but does not change relative risk of fortifier type.
- Impact of Study Design Differences on Results: Some included studies had different baseline diets (e.g., use of donor milk vs. preterm formula as supplementation). This could introduce significant heterogeneity. Was any stratified analysis conducted to address this potential confounder?
Thank you for your comment. Unfortunately, stratified analyses aren’t feasible because of the network structure. Analyses restricting to a smaller network is not methodologically robust. __https://bmcmedresmethodol.biomedcentral.com/articles/10.1186/s12874-023-01959-9.
We did address this concern as well as we could through trying to balance splitting/lumping nodes together and we highlighted this in the limitations as a potential source of heterogeneity/intransitivity.
Round 2
Reviewer 1 Report
Comments and Suggestions for Authors
Accept in present form
Reviewer 2 Report
Comments and Suggestions for Authors
Thank you very much for allowing me to review the manuscript “nutrients-3552039_Effect of multi-nutrient milk fortification on preterm neonate outcomes: a network meta-analysis” once again. I also wish to express my appreciation for the effort the authors have made to clarify the previous concerns and to provide the requested information.
Comments:
If the study is presented in the title as a network meta-analysis (NMA), this should be consistently reflected throughout the manuscript. Alternative study designs or terminologies should not be used. Therefore, the authors should ensure consistency in the designation of the study type across the entire document.
Regarding the abstract, when clarification was requested about the review period, the intention was not to include a specific date (day), but rather to define the review period with clear start and end points—specifically, the month and year when the review began and when it concluded. This information is essential to understand the timeframe covered by the review. Additionally, the results section should include not only qualitative but also quantitative data.
The conclusion presented in the abstract does not adequately address the study’s objective. While it states that this is the first comprehensive study to evaluate fortification feeding and highlights the lack of sufficient evidence, it does not provide a direct response to the main objective of the review.
Moreover, the manuscript does not explain why the review ends in 2023, despite the current year being 2025. Such an explanation is necessary to justify the time frame of the review.
All research papers, whether original studies or reviews, are generally based on a hypothesis. Thus, it is unclear why this manuscript does not begin with a clearly stated hypothesis. A possible hypothesis could involve the comparison of fortification strategies, for instance.
In the Materials and Methods section, while the PRISMA diagram indicates the number of studies identified and excluded, it does not specify the inclusion and exclusion criteria applied. Clearly stating these criteria is a standard part of the methodological reporting and should be included. Furthermore, although line 120 mentions that PICO criteria were used, these criteria are not detailed anywhere in the manuscript.
Author Response
Nutrients-3552039_Effect of multi-nutrient milk fortification on preterm neonate outcomes: a network meta-analysis”
Thank you for your additional comments, which we have addressed below.
Response to reviewer 2-Round 2
If the study is presented in the title as a network meta-analysis (NMA), this should be consistently reflected throughout the manuscript. Alternative study designs or terminologies should not be used. Therefore, the authors should ensure consistency in the designation of the study type across the entire document.
Thank you for your comment. As we mentioned previously, we have completed a thorough review of the manuscript use the term NMA consistently throughout the paper in relation to our study design.
Regarding the abstract, when clarification was requested about the review period, the intention was not to include a specific date (day), but rather to define the review period with clear start and end points—specifically, the month and year when the review began and when it concluded. This information is essential to understand the timeframe covered by the review. Additionally, the results section should include not only qualitative but also quantitative data.
Thank you we have corrected this as requested. The search was conducted from up to June 23, 2023. There was no start date, we took all papers published up to June 27, 2023. Abstract and Line 84 revised manuscript.
The conclusion presented in the abstract does not adequately address the study’s objective. While it states that this is the first comprehensive study to evaluate fortification feeding and highlights the lack of sufficient evidence, it does not provide a direct response to the main objective of the review.
We believe that our conclusion now reflects our hypothesis and study aim.
Moreover, the manuscript does not explain why the review ends in 2023, despite the current year being 2025. Such an explanation is necessary to justify the time frame of the review.
The workload regarding the conduct of such an extensive NMA is significant. Once the search is performed, it takes months of work to determine eligibility, screening, data extraction, follow up with authors even before analysis is performed. Analysis is complex and takes considerable hours of work to ensure accuracy. We recognize the date of the search but also discuss that to our knowledge not other RCT comparing fortification practices on infant outcomes has been completed. As such, the results reflect the current existing literature.
All research papers, whether original studies or reviews, are generally based on a hypothesis. Thus, it is unclear why this manuscript does not begin with a clearly stated hypothesis. A possible hypothesis could involve the comparison of fortification strategies, for instance.
We have included a hypotheses statement in the revised manuscript.
In the Materials and Methods section, while the PRISMA diagram indicates the number of studies identified and excluded, it does not specify the inclusion and exclusion criteria applied. Clearly stating these criteria is a standard part of the methodological reporting and should be included. Furthermore, although line 120 mentions that PICO criteria were used, these criteria are not detailed anywhere in the manuscript.
Thank you. The following information is included under methods, which we believe addresses the reviewers comment.
“The population of interest was preterm neonates defined as those delivered less than 37 weeks gestational age cared for in a neonatal intensive care unit (NICU). The intervention of interest was fortification of milk with the intention of increasing caloric or nutrient intake. Single amino acids, DHA, and probiotics supplementation alone did not qualify. To be eligible for inclusion, randomized clinical trials must have reported on at least one primary outcome and included at least one arm with human milk. Any neonatal diet was included as a comparator including exclusive formula diets.” Lines 92-98.